# Online Learning with Multiple Fairness Regularizers via Graph-Structured Feedback

**Quan Zhou**[*]                                         *quan.zhou@nus.edu.sg*
*Department of Mathematics, National University of Singapore*

**Jakub Mareček**                                        *jakub@marecek.cz*
*Department of Computer Science, Czech Technical University*

**Robert Shorten**                                       *r.shorten@imperial.ac.uk*
*Dyson School of Design Engineering, Imperial College London*

**Reviewed on OpenReview:** *https://openreview.net/forum?id=y8iWuDZtEw*

## Abstract

There is an increasing need to enforce multiple, often competing, measures of fairness within automated decision systems. The appropriate weighting of these fairness objectives is typically unknown a priori, may change over time and, in our setting, must be learned adaptively through sequential interactions. In this work, we address this challenge in a bandit setting, where decisions are made with graph-structured feedback.

## 1 Introduction

Artificial intelligence (AI) has become deeply integrated into modern society, powering applications across business, healthcare, finance, and public policy. Its rapid adoption stems from its ability to automate complex decision-making processes at scales previously unimaginable. However, as AI systems increasingly influence outcomes that directly affect individuals and communities, ensuring fairness becomes a critical concern. This work focuses specifically on fairness in bandit-based sequential decision-making—settings where decisions must be made repeatedly with only partial, stochastic feedback. Fairness in such sequential AI systems is not only a matter of ethical responsibility but also a practical necessity for building trust, meeting regulatory standards, and supporting long-term social and organizational goals.

One of the main challenges in achieving fairness is that it can be defined in numerous, often conflicting, ways. Measures of fairness can focus on individuals, subgroups, or combinations of protected attributes such as race and gender. Optimizing for one notion of fairness frequently comes at the expense of another, and fairness objectives can also conflict with traditional performance metrics. As a result, trade-offs are inherent in the design of fair AI systems. These trade-offs must be incorporated into decision-making. Yet, the structure of fairness–fairness and fairness–performance trade-offs is often difficult to characterize analytically. In the long term, societal preferences and definitions of fairness are not static but evolve with changing norms, values, and data distributions. This dynamic nature introduces further complexity to fairness in AI.

Consider, for example, political advertising on the Internet. In some jurisdictions, the current and proposed regulations of political advertising suggest that the aggregate space-time available to each political party should be equalised in the spirit of "equal opportunity" [1]. It is not clear, however, whether the opportunity should be construed in terms of budgets, average price per ad, views, "reach" of the ad, and whether it should correspond to the share of the popular vote in past elections (e.g., in a previous election with some means of data imputation), the current estimates of voting preferences (cf. rulings in the US allowing for

---

[*]Work done while a PhD student at Imperial College London.
[1]In the USA, see the equal opportunity section (315) of the Communications Act of 1934, as amended many times.

the participation of only the two leading parties), or be uniform across all registered parties (cf. the "equal time" rule in broadcast media). For example, on Facebook, the advertisers need to declare their affiliation with the political party they support[2]. Considering multiple political parties, as per the declaration, and their budgets, one may need to consider multiple fairness measures (e.g., differences in average price per ad, differences in spend proportional to the vote share, differences in the reach, etc, and their $l_1, l_2, l_\infty$ norms). Furthermore, most platforms have a very clear measure of ad revenue, which they wish to balance with some "fairness regularizers" Lu et al. (2020). To complicate matters further still, the perceptions of the ideal trade-off among the ad revenue and the fairness regularizers clearly change over time.

In this work, we propose to address the challenge of conflicting fairness notions by explicitly modeling and exploring the space of trade-offs. Fairness measures are often interdependent: some may conflict, while others may partially align, and decisions made at each time step can improve one fairness criterion while negatively impacting another. To capture these relationships, we represent fairness regularizers and available actions as nodes in a graph, with edges encoding their interactions, which can be either time-invariant or time-varying to account for evolving social norms. Building on graph-structured bandits (Mannor & Shamir, 2011), we integrate this representation into a sequential decision-making framework, enabling sampling-based exploration of the feasible fairness landscape. This approach provides a practical way for decision-makers to navigate trade-offs intelligently, balancing multiple fairness objectives.

**Contributions:** (1) We formulate the problem of learning under time-varying, multiple fairness constraints as an online resource allocation problem with graph-structured partial feedback. (2) We develop a bandit algorithm for this setting by extending the graph-structured bandits framework: we model fairness regularizers as additional nodes in the feedback graph, which classically only encodes relationships among actions. This enables sequential decisions that balance reward and non-stationary fairness goals.

Our work is inspired by Awasthi et al. (2020), which utilizes an incompatibility graph among multiple fairness regularizers (criteria) and assumes full feedback from all regularizers. Our work differs from prior approaches that formulate fairness-aware sequential decision-making as a Markov Decision Process (MDP). Whereas MDP-based methods typically assume full state observability and known transition dynamics, we restrict attention to the bandit setting, where the learner observes only stochastic rewards from actions, and correlations among actions are embedded in the graph-structured feedback. This makes our framework more suitable for applications where system dynamics are unknown, non-stationary, or too costly to model explicitly.

## 2 Related Work

There are many measures of fairness for any protected attributes (e.g., gender, race, ethnicity, income) defining the subgroups. Without attempting to encompass all of the related literature, let us present some of the most relevant work.

### 2.1 Measures of Fairness

There are many definitions of fairness, especially within the context of classification. The statistical definition of fairness requires that a classifier's statistics—such as the raw positive classification rate (also sometimes referred to as statistical parity), false positive rate, and false negative rate (also sometimes referred to as equalized odds)—be equalized across subgroups so that the algorithm's errors are proportionately distributed. An initial, naive approach to fairness is to enforce "fairness through unawareness" by excluding protected attributes from the decision model. The shortcoming of this notion is that there may exist unobserved features related to protected attributes, and these features can be used to predict the protected attributes. Therefore, a predictor that ignores protected attributes can still produce outcomes correlated with them. Demographic parity, proposed by Calder et al. (2009), requires that members of each segment of a protected class (e.g., gender) receive the positive outcome at equal rates. However, this criterion may lead to unfairness when the distributions of features differ between advantaged and disadvantaged subgroups, even in the

---

[2]In the USA, this may become a legal requirement, cf. the Honest Ads Act. Cf. `https://www.congress.gov/bill/115th-congress/senate-bill/1989`

absence of bias. The notions of equalized odds introduced by Hardt et al. (2016) and counterfactual fairness proposed by Kusner et al. (2017) both require the predictor to be unrelated to protected attributes. In other words, these approaches can be viewed as refined versions of unawareness that aim to remove the effects of unobserved features correlated with protected attributes. They focus on ensuring accurate prediction in the presence of unbalanced distributions, without introducing discrimination. The underlying belief is that a predictor is unlikely to be discriminatory if it only reflects real outcomes.

Group fairness provides only an average guarantee for individuals within a protected group Awasthi et al. (2020) and is insufficient on its own. Even when group fairness criteria are satisfied, outcomes may still be unfair from the perspective of individual members. Individual fairness requires imposing constraints on specific pairs of individuals, rather than on quantities averaged over groups. In other words, it requires that similar individuals should be treated similarly Dwork et al. (2011). This notion relies on the existence of a similarity metric that captures ground truth, which typically requires both general and task-specific agreement on its definition.

## 2.2 The Conflict Between Group and Individual Fairness

In the United States, affirmative action policies are often controversial because favoring one group inevitably involves disadvantaging another Dur et al. (2020). In the case Regents of the University of California v. Bakke (1978), race was allowed to be one of several factors in college admission decisions. However, California Proposition 209 later prohibited state governmental institutions from considering race, sex, or ethnicity in public employment, public contracting, and public education. In 2019, California Senate Constitutional Amendment No. 5 (SCA-5) sought to eliminate Proposition 209's ban on the use of race, sex, and other characteristics. Asian American communities opposed this amendment Wang (2020). These debates reflect the collision of different perceptions of fairness, which stem from distinct views regarding which factors of individuals' performance they should be held accountable for Schildberg-Hörisch et al. (2020).

## 2.3 Reasoning about Trade-offs

Recently, there have been several attempts to formulate frameworks for reasoning about multiple fairness measures. The "fairness resolution model" proposed by Awasthi et al. (2020) is guided by unfairness complaints received by the system and offers a more practical way to maintain both group and individual fairness. This work provides a finite-state, discrete-time Markov decision process framework that supports multiple fairness regularizers while accounting for their potential incompatibilities. Independently, Ospina et al. (2021) proposed a framework that employs an online algorithm for time-varying networked system optimization, aiming to trade off human preferences. In particular, the human preference function (i.e., the fairness regularizer function) is learned concurrently with the execution of the algorithm using shape-constrained Gaussian processes. Other works have focused on designing systems that balance multiple, potentially conflicting fairness measures (Kim et al., 2020; Lohia et al., 2019), and on specifically identifying a Pareto front of fairness–performance trade-offs Kozdoba et al. (2024).

Prior research has emphasized that naively enforcing fixed fairness constraints can sometimes have unintended negative consequences, such as harming minority groups through delayed or compounding effects (Liu et al., 2018c; D'Amour et al., 2020). To address this, researchers have explored approaches that allow AI models to adapt to evolving fairness notions and shifting environments. In particular, sequential decision-making frameworks have incorporated mechanisms for feedback and long-term outcome monitoring, enabling policies to adjust dynamically as fairness concepts and societal expectations change (Bechavod & Roth, 2023; Wen et al., 2021; Zhang & Liu, 2021; Jabbari et al., 2017).

Alternatively, one can consider approaches from multi-objective optimization (MOO). A key recent reference is Zhang et al. (2021), who study dynamic MOO but do not provide performance guarantees for their algorithms. This line of work builds on a long history of research on convexification techniques in MOO (e.g., Sun et al., 2001). Related developments have also appeared in various applied domains, such as mathematical finance Li & Ng (2000).

| Algorithm | Reference | Regret (allowing for minor variations) |
|---|---|---|
| ELP | Mannor & Shamir (2011) | |
| ExpBan | Mannor & Shamir (2011) | $\mathcal{O}(\sqrt{\bar{\chi}(G)\log(k)T})$ |
| Exp3-Set | Alon et al. (2013) | $\tilde{\mathcal{O}}(\sqrt{\alpha T \log d})$ |
| UCB-N | Caron et al. (2012) | Expected regret parametrised by clique covers |
| Exp3-Dom | Alon et al. (2013) | |
| Exp3-IX | Kocák et al. (2014) | Stochastic feedback graph |
| UCB-LP | Buccapatnam et al. (2014) | Stochastic feedback graph |
| Exp3.G | Alon et al. (2015) | Tight bounds for some cases |
| Exp3-WIX | Kocák et al. (2016) | $\tilde{\mathcal{O}}(\sqrt{\alpha^* T})$ |
| CHK | Cohen et al. (2016) | Tight bounds for some cases matching Alon et al. (2015) |
| BARE | Carpentier & Valko (2016) | |
| Exp3-DOM | Alon et al. (2017) | $\mathcal{O}\left(\log(K)\sqrt{\log(KT)\sum_{t\in[T]}\mathrm{mas}(\mathcal{G}^t)}\right)$ |
| ELP.P | Alon et al. (2017) | $\mathcal{O}\left(\sqrt{\log(K/\delta)\sum_{t\in[T]}\mathrm{mas}(\mathcal{G}^t)}\right)$ w.p. $1-\delta$ |
| TS | Tossou et al. (2017) | |
| TS | Liu et al. (2018b) | Optimal bayesian regret bounds |
| OMD | Arora et al. (2019) | Switching costs |
| TS+UCB | Lykouris et al. (2020) | |
| IDS | Liu et al. (2018a) | |
| UCB-NE | Hu et al. (2020) | Non-directed graphs |
| UCB-DSG | Cortes et al. (2020) | Pseudo-regret bounds |
| | Li et al. (2020) | Cascades in the stochastic f.g. |
| OSMDE | Chen et al. (2021) | $\mathcal{O}((\delta^*\log(|V|))^{1/3}T^{2/3})$ |
| | Lu et al. (2021) | Adversarial corruptions |

Table 1: An overview of the algorithms for the model with graph-structured feedback. $\mathrm{mas}(\mathcal{G}^t)$ is the size of the maximal acyclic graph in $\mathcal{G}^t$, $\chi$ is the coloring number, $\delta^*$ is the weak domination number, $\alpha$ is the independence number, $\alpha^*$ is the effective independence number.

## 2.4 Graph-Structured Bandits

Graph-structured bandits (Alon et al., 2013; Mannor & Shamir, 2011) generalize the classical multi-armed bandit setting (Lai & Robbins, 1985) by incorporating a known dependency structure among the actions (arms), typically represented as a graph. In this formulation, each vertex corresponds to an action, and edges encode relationships such as similarity, correlation, or shared feedback. Executing an action not only yields a reward for that action but may also provide partial feedback about its neighbors, enabling more efficient exploration. This structure allows algorithms to leverage side information and propagate observed rewards across the graph, reducing uncertainty and improving regret bounds compared to standard bandit approaches. This approach provides a middle ground between the classical bandit setting and the expert setting. In the standard bandit setting, only the reward of the chosen vertex is revealed; the best-known algorithm Audibert et al. (2009) achieves optimal regret of order $\sqrt{|A|T}$. In contrast, in the expert setting, the rewards of all vertices are revealed, and the optimal regret of order $\sqrt{\log(|A|)T}$ can be achieved using the Hedge algorithm Freund & Schapire (1997) or the Follow the Perturbed Leader algorithm Kalai & Vempala (2005).

Graph-structured bandits naturally apply to scenarios where dependencies among actions exist, including recommendation systems, networked decision-making, and, as we explore in this paper, reasoning about multiple fairness measures in sequential decision problems. In this work, we first formulate the problem of handling multiple fairness measures as a graph-structured bandits problem. Once this framework is established, we present algorithms suitable for this setting. In the context of graph-structured feedback, numerous algorithms have been developed, as summarized in Table 1.

# 3 Our Framework for Reasoning with Multiple Fairness Regularizers

As the need for additional notions of model fairness grows and the trade-offs among them become increasingly important, it is necessary to develop a comprehensive framework for reasoning about multiple fairness regularizers—particularly in cases where not all regularizers can be fully satisfied simultaneously.

We consider discrete time, with $T$ denoting the number of time steps (rounds). We assume the existence of a possibly time-varying, (un)directed compatibility graph $\mathcal{G}^t = (A, F, E^t)$. The vertex set includes action vertices $a \in A$ and fairness regularizer vertices $f \in F$. The (possibly time-varying) relationships among them are encoded in the edge set $E^t$, where $v \xrightarrow{t} v'$ denotes an edge from vertex $v$ to vertex $v'$ at time $t$, with $v \in A \cup F$. Specifically, there are two types of relationships:

$a \xrightarrow{t} f$: A directed edge from an action vertex $a$ to a regularizer vertex $f$ indicates that executing action $a$ will affect regularizer $f$. This relation is captured by the subset of action vertices that can influence $f$ in round $t$.

$a \xrightarrow{t} a'$: A directed edge from an action vertex $a$ to another action vertex $a'$ indicates that executing action $a$ will reveal the reward of executing action $a'$. This relation is captured by the subset of action vertices whose rewards will be disclosed when $a$ is executed in round $t$.

In fact, if we remove the set of regulariser vertices $F$, the graph reduces to the standard graph-structured bandit setting. The introduction of the extra vertex set $F$ allows us to explicitly account for the states of the fairness regularisers and their associated rewards.

Suppose there exists a state space $S$. Each regulariser vertex $f \in F$ is characterized by a state $s \in S$. The state of a regulariser vertex evolves whenever any action vertex $a$ such that $a \xrightarrow{t} f$ are executed at time $t$, as formalized in equation 1:

$$s^{(f,t)} = \begin{cases} P^t\big(s^{(f,t-1)}, a^t\big) & \text{if } a^t \xrightarrow{t} f, \\ s^{(f,t-1)} & \text{otherwise,} \end{cases} \tag{1}$$

where $P^t$ is the state evolution function.

The reward $r^t$ at round $t$ consists of the direct income from executing an action and penalties imposed by the fairness regularisers. Formally,

$$r^t(a) := \text{Income}^t(a) - \sum_{f \in F} f\big(s^{(f,t)}\big), \tag{2}$$

where $\text{Income}^t(a)$ denotes the direct reward from executing action $a$, and $f(s^t)$ represents the penalty associated with regulariser $f$ in round $t$, which is a function of the states $s^t$.

Within this general framework, we consider the case of limited graph-structured feedback. While the functions $P^t$ and $\text{Income}^t$ are unknown, the compatibility graph $\mathcal{G}^t$ is provided at the beginning of each round and may be time-invariant or time-varying. (For the case where the graph is time-varying and not disclosed, we refer to Algorithm 1 in Alon et al. (2017).) In each round, we choose an action $a$ and observe the rewards associated with action vertices $a'$ such that $a \xrightarrow{t} a'$ according to $\mathcal{G}^t$. This limited feedback is then used to guide decisions in subsequent rounds.

Notice that in the limited-feedback setting, Awasthi et al. (2020) introduced a compatibility graph consisting only of regulariser vertices in the context of fairness. Their approach considers a sequence of fairness rewards, which are functions of the regulariser vertices' states and are received at each time step. These rewards reflect users' perceptions of fairness regarding the outcomes. However, we can also incorporate the actual fairness outcomes as feedback, since there may be a discrepancy between users' perceptions and reality.

Suppose Algorithm Alg is used to select $a^t$ in each round. The overall reward of the algorithm is defined as $R(\mathrm{Alg}) := \sum_{t \in [T]} r^t(a^t)$. We consider two types of regrets:

- **Dynamic regret:** the difference between the cumulative reward of algorithm and that of the best sequence of actions chosen in hindsight, thus $\mathrm{OPT}_D - R(\mathrm{Alg})$, where

$$\mathrm{OPT}_D = \sum_{t \in [T]} \max_{a \in V} r^t(a). \tag{3}$$

- **Weak regret:** the difference between the cumulative reward of algorithm and that of the best single action chosen in hindsight, applied at all time steps, thus $\mathrm{OPT}_W - R(\mathrm{Alg})$, where

$$\mathrm{OPT}_W = \max_{a \in V} \sum_{t \in [T]} r^t(a). \tag{4}$$

In Table 2, we provide a summary of the common notation used in the paper.

| Symbol | Description |
|---|---|
| $t, T$ | Round index, total number of rounds |
| $k$ | (Political) party index |
| $A$ | Set of action vertices |
| $F$ | Set of fairness regularizer vertices |
| $E^t$ | Edge set of the feedback graph |
| $\mathcal{G}^t = (A, F, E^t)$ | Time-varying compatibility graph at round $t$ |
| $\mathrm{mas}(\mathcal{G})$ | Size of the maximal acyclic graph in $\mathcal{G}$ |
| $a \xrightarrow{t} f$ | Directed edge from $a \in A$ to $f \in F$ in $\mathcal{G}^t$ |
| $a \xrightarrow{t} a'$ | Directed edge from $a \in A$ to $a' \in A$ in $\mathcal{G}^t$ |
| $a^t$ | Action chosen at round $t$ |
| $s^{(f,t)}$ | State of fairness regularizer $f$ after round $t$ |
| $f(s^{(f,t)})$ | Penalty (negative reward) of fairness regularizer $f$ at round $t$ |
| $\mathrm{Income}^t$ | Income (reward) vector at round $t$ |
| Share | Target shares for each party (fairness goal) |
| $R(\mathrm{Alg})$ | Cumulative reward of algorithm Alg over $T$ rounds |
| $\mathrm{OPT}_W$ | Optimal cumulative reward of the best sequence of actions in hindsight |
| $\mathrm{OPT}_D$ | Optimal cumulative reward the best single in hindsight |
| $\eta$ | Learning rate |
| $\delta$ | Confidence parameter |
| $[n]$ | Set $\{1, 2, \ldots, n\}$ |

Table 2: Summary of common notation used in the paper.

## 4 Motivating Examples

Let us revisit the example of political advertising on the Internet, which we introduced earlier. Consider two major political parties—e.g., the Conservative and Liberal parties—along with several third-party candidates ($k = 1, 2, 3$) in a jurisdiction where political advertisements on social networking sites are regulated.

Figure 1(a) provides a general example of the compatibility graph, featuring three action vertices, three regulariser vertices, and nine subgroup regulariser vertices. According to the edge definitions, executing action vertex $a^1$ could potentially affect $f_1, f_1^1, f_1^2, f_1^3$, and the rewards associated with $a^2$ and $a^3$ would be disclosed if $a^1$ is executed.

This setup models a scenario in which the regularisers of political advertising represent dollar spent, reach, and number of shares, respectively, each of which is required to satisfy an "equal opportunity" constraint.

The subgroup regulariser vertices $f_k^1, f_k^2, f_k^3$ impose constraints on the same metrics (dollar spent, reach, number of shares) for party $k$, ensuring they remain within specific ranges. While there could be many more actions in practice, in this example we consider $a_1, a_2, a_3$ as selling one unit of advertising space-time to the Conservative party, the Liberal party, and the third-party candidates, respectively.

As another example, consider the same three political parties. There are three action vertices ($A = \{a_1, a_2, a_3\}$) and three regulariser vertices ($F = \{f_1, f_2, f_3\}$), each corresponding to one party. The action $a_k$ represents selling one unit of advertising space-time to party $k$. The state $s^{(f_k,t)}$ represents the cumulative number of space-time units sold to party $k$ up to round $t$, and we assume the initial states of the regulariser vertices are $[0, 0, 0]$.

Suppose the regularisers suggest that the allocation $s^{(f_k,t)}{}_{k=1,2,3}$ should be proportional to the share of the popular vote. One could then set the regularisers $f_1, f_2, f_3$ to require the two major parties and all third-party candidates to receive an equalized allocation of political ads on a social media platform, i.e., $s^{(f_k,t)}/T = 1/3$ for $k = 1, 2, 3$, In each round, each regulariser $f_k$ returns a penalty based on how far the allocation for party $k$ deviates from its target share: $f_k(s^{(f_k,t)}) = \left| \frac{s^{(f_k,t)}}{T} - \frac{1}{3} \right|$. Let Income$^t$ be the income vector at time $t$. The income from executing action $a_k$ in round $t$ is denoted Income$^t(a_k)$, i.e., the $k$-th component of Income$^t$.

Let us consider $T = 2$, and the income vector for the two rounds to be:

$$\text{Income}^1 = \begin{bmatrix} 1 & 0 & 0 \end{bmatrix}, \text{Income}^2 = \begin{bmatrix} 0 & 1 & 0 \end{bmatrix}.$$

Here, we would like to learn the trade-off between the regularizers and the income of the platform. Correspondingly, the state transmission function $P$ is set to be $s^{(f_k,t)} = s^{(f_k,t-1)} + 1$ if $a_k$ is executed in time $t$ otherwise the state stays the same.

**OPT$_D$:** In the example above, the best sequence of actions would choose different actions of each round. Perhaps it could conduct $a_1$ in the first round and $a_2$ in the second round. Let $s^{(F,t)}$ be the 3-dimensional vector $[s^{(f_1,t)}, s^{(f_2,t)}, s^{(f_3,t)}]$, then with the corresponding states being

$$s^{(F,1)} = \begin{bmatrix} 1 & 0 & 0 \end{bmatrix}, \quad s^{(F,2)} = \begin{bmatrix} 1 & 1 & 0 \end{bmatrix}.$$

The resulting reward is

$$\text{OPT}_D = 2 - \left( \left| 1 - \frac{1}{3} \right| + 2 \times \frac{1}{3} + 2 \times \left| \frac{1}{2} - \frac{1}{3} \right| + \frac{1}{3} \right).$$

**OPT$_W$:** If, on the other hand, we were to pick only a single action vertex to be taken in both rounds $t$. The corresponding actions and states would be:

$$s^{(F,1)} = \begin{bmatrix} 1 & 0 & 0 \end{bmatrix}, \quad s^{(F,2)} = \begin{bmatrix} 2 & 0 & 0 \end{bmatrix}.$$

The resulting reward is

$$\text{OPT}_W = 2 - 2 \times \left( \left| 1 - \frac{1}{3} \right| + 2 \times \frac{1}{3} \right).$$

**Limited Feedback:** When $P^t$ and Income$^t$ are unknown and only the rewards $r^t$ are revealed, our feedback at time $t$ is therefore limited to: the reward for the chosen action $a^t$, plus the rewards for any action $a$ where $a^t \xrightarrow{t} a$ in the compatibility graph $\mathcal{G}^t$.

Figure 1(c) provides an example of a time-invariant graph, which must be disclosed before the first round. According to the edge definitions, if $(a_2 \xrightarrow{t} a_3)$ and we select action $a_2$ in round $t$, the reward $r^t(a_2)$ that we actually achieve will be revealed immediately. Simultaneously, we also observe the reward $r^t(a_3)$ that we could have obtained had we selected action vertex $a_3$.

The compatibility graph $\mathcal{G}^t$ can also be time-varying, in which case it is disclosed at the beginning of each round $t$. This can model situations such as temporary malfunctions or availability constraints for certain actions.

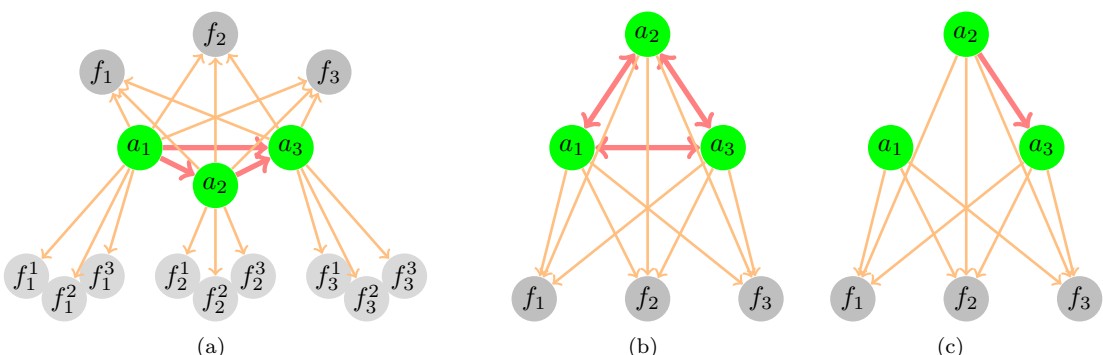

Figure 1: Examples of the compatibility graph with 3 action vertices (green), i.e., $a_1, a_2, a_3$ and 3 regularizer vertices (dark grey), i.e., $f_1, f_2, f_3$. Additionally, (a) has extra subgroup regularizer vertices (light grey), i.e., $f_k^1, f_k^2, f_k^3$ for each subgroup $k \in [1, 2, 3]$.

## 5 A Model with Graph-Structured Bandit Feedback

Starting with Mannor & Shamir (2011), this line of work introduced a graphical feedback system to model the side information available in online learning. In the graph (corresponding to the action vertices and edges among them in our framework), a directed edge from vertex $a$ to vertex $a'$ implies that by choosing $a$, the rewards associated with both $a$ and $a'$ are revealed immediately. Note that an undirected edge between vertices $a$ and $a'$ can be interpreted as two directed edges, $a \xrightarrow{t} a'$ and $a' \xrightarrow{t} a$.

### 5.1 Time-variant Graph-Structured Bandit Feedback

There are several algorithms available in the literature for graph-structured feedback. Mannor & Shamir (2011) introduced the ExpBan and ELP algorithms, where ELP allows for time-varying graphs, as detailed in Algorithm 1. When the graph is disclosed only after the action is taken, the Exp3-SET algorithm suffices, with complete analysis provided in Alon et al. (2015), which derives regret bounds for both observable and unobservable graphs. In the context of multiple fairness regularisers, we model the compatibility graph as a directed, informed, and non-stochastic variant, following Alon et al. (2017).

In this setting, the reward function $r^t$ in round $t$ can be an arbitrary bounded function of the history of actions. The compatibility graph may be time-varying but is disclosed at the beginning of each round. Immediately after choosing action $a^t$, the reward associated with $a^t$ is revealed, as well as the rewards associated with other actions $a$ if $a^t \xrightarrow{t} a$.

Let $\Delta(A)$ denote the probability simplex over the set of action vertices $A$. Let $q^{(i,t)}$ be the probability of observing the reward of action vertex $a^i$ in round $t$, and $p^{(i,t)}$ the probability of selecting $a^i$ in that round. In Algorithm 1, $p^{(i,t)}$ is determined as a trade-off between the weight $\omega^{(i,t)}$ and the exploration factor $\xi^{(i)}$, modulated by an egalitarianism factor $\gamma^t$. Here, $\omega^{(i,t)}$ represents the weight of each action vertex, which increases (decreases) when the associated payoff is favorable (unfavorable), while $\xi^{(i)}$ in equation 5 represents the desire to select an action vertex uniformly from each clique of action vertices in $\mathcal{G}^t$. Using this setup, the importance sampling estimator $\hat{r}^{(i,t)}$ and the weight $\omega^{(i,t)}$ for each $i \in A$ can be updated as described in (6–7).

**Theorem 1** (Informal version). *Algorithm 1 achieves weak regret bounded by*

$$\widetilde{O}\left(\sqrt{\log(|A|/\delta) \sum_{t \in [T]} mas(\mathcal{G}_A^t)}\right),$$

*where $\widetilde{O}$ hides only numerical constants and factors logarithmic in $|A|$ and $1/\eta$, and $mas(\mathcal{G}_A^t)$ denotes the size of a maximum acyclic subgraph of the action feedback graph $(A, E^t)$.*

---

**Algorithm 1** Exponentially-Weighted Algorithm with Linear Programming

---

**Input**: Action vertices $A$, regularizer vertices $F$ and rounds $T$, compatibility graph $\mathcal{G}^t$ in each round (or $\mathcal{G}$ if time-invariant), confidence parameter $\delta \in (0,1)$, learning rate $\eta \in (0, 1/(3|A|)]$.
**Output**: Actions and states.
**Initialization**: $\omega^{(a,1)} = 1$, for $a \in A$.

1: **for** $t = 1, \ldots, T$ **do**
2:      Solve the linear program

$$\max_{\xi \in \Delta(A)} \min_{a \in A} \sum_{a': a' \xrightarrow{t} a} \xi^{(a')} \tag{5}$$

3:      Set $p^{(a,t)} := (1 - \gamma^t)\,\omega^{(a,t)}/W^t + \gamma^t\,\xi^{(a)}$, where $W^t = \sum_{a \in A} \omega^{(a,t)}$,

$$\gamma^t = \frac{(1+\beta)\eta}{\min_{a \in A} \sum_{a': a' \xrightarrow{t} a} \xi^{(a')}}, \quad \beta = 2\eta \sqrt{\frac{\ln(5|A|/\delta)}{\ln|A|}}.$$

4:      Update $q^{(a,t)} = \sum_{a': a' \xrightarrow{t} a} p^{(a',t)}$.
5:      Draw one action vertex $a^t$ according to distribution $p^{(a,t)}$.
6:      Observe pairs $r^{(a,t)}$ for all actions $a$ such that $a^t \xrightarrow{t} a$, where $r^{(a,t)}$ is given as equation 2.
7:      For any $a \in A$, set estimated reward $\hat{r}^{(a,t)}$ and update $\omega^{(a,t+1)}$, as follows

$$\hat{r}^{(a,t)} = \frac{r^{(a,t)} \mathbf{1}\{a^t \xrightarrow{t} a\} + \beta}{q^{(a,t)}}, \tag{6}$$

$$\omega^{(a,t+1)} = \omega^{(a,t)} \exp\left(\eta\,\hat{r}^{(a,t)}\right). \tag{7}$$

8: **end for**

---

As summarized in Alon et al. (2013), the regret bounds in graph-structured bandits interpolate between the bandit and full-information settings. The size of the maximum acyclic subgraph $\mathrm{mas}(\mathcal{G}_A^t)$ is the largest possible subgraph of a directed graph that contains no directed cycles. It quantifies the information complexity of the feedback structure: a larger $\mathrm{mas}(\mathcal{G}_A^t)$ implies a denser, more informative feedback pattern. When $(A, E^t)$ is a complete graph for all $t$ – meaning the decision-maker observes the losses of all actions, as in the full-information setting – we have $\mathrm{mas}(\mathcal{G}_A^t) = 1$. In this case, this bound recovers the standard $\widetilde{O}(\sqrt{T})$ regret (up to logarithmic factors). Conversely, when $G^t$ is an empty graph for all $t$ – meaning the decision-maker only observes the loss of the chosen action, as in the bandit setting – then $\mathrm{mas}(\mathcal{G}_A^t) = |A|$, and we recover the standard $\widetilde{O}(\sqrt{|A|T})$ bandit regret. Between these extremes, the regret bound scales with the graph structure of the action vertices, as captured by the term $\sum_{t \in [T]} \mathrm{mas}(\mathcal{G}_A^t)$. The formal version of the theorem and proof is included in the Supplementary Material, but essentially follows from the work of Alon et al. (2017); Mannor & Shamir (2011).

## 6 Numerical Illustrations

Let us revisit and extend the motivating example in Section 4 by considering five political parties, indexed by $k = 1, \ldots, 5$. We define five action vertices $A = \{a_1, \ldots, a_5\}$, where $a_k$ corresponds to selling one unit of advertising space-time to party $k$'s candidates. The state variable $s^{(f_k,t)}$ represents the cumulative number of space-time units sold to party $k$ up to round $t$.

We also introduce five regularizers $F = \{f_1, \ldots, f_5\}$, each requiring that party $k$'s candidates receive a prescribed share of the total advertising space-time—for example, to match the most recent election results. The target shares are sampled from a Dirichlet process with concentration parameter $\alpha = 1$ and mean equal to the uniform distribution. Let $\mathrm{Share}_k$ denote the target share for party $k$. The corresponding regularizer

penalizes deviations from this target:

$$f_k\big(s^{(f_k,t)}\big) = \big|s^{(f_k,t)}/t - \mathrm{Share}_k\big|,$$

for each $t \in [T]$, where the penalty is a negative reward. The state transmission function is defined as $s^{(a,t)} = s^{(a,t-1)} + 1$ if action $a$ is chosen at round $t$, and $s^{(a,t)} = s^{(a,t-1)}$ otherwise. The initial states for all parties are zero.

The platform's revenue at time $t$ is $\mathrm{Income}^t(a^t)$, given by the $a^t$-th component of the income vector $\mathrm{Income}^t$. We evaluate our method using revenue data from the 5 advertisers (parties) over 100,000 impressions—forming a $5 \times 100{,}000$ income matrix—drawn from the dataset of Balseiro et al. (2020). For each row of this matrix, we first normalize its entries to the range $[0, 1]$ and then add 1, ensuring that the net reward at time $t$—i.e., income minus penalty—is nonnegative.

For Algorithm 1, we set the learning rate $\eta = 1/15$ and the confidence parameter $\delta = 0.025$. The time horizon $T$ is varied over $[30, 80]$ in increments of 5. For each $T$, we run 30 repeated trials, each selecting $T$ rows of revenue data uniformly at random from the income matrix. We compare three experimental cases:

- **Baseline (empty graph).** No edges between action vertices, corresponding to a classic bandit setting where only the reward of the chosen action is observed.

- **Time-invariant graph.** A single randomly generated compatibility graph is used for all rounds in each trial.

- **Time-varying graph.** A new compatibility graph is randomly sampled in each round.

Note that when we generate a random graph, we refer specifically to the action feedback graph among action vertices, since the edges between actions and regularizers are fixed by the definition of the vertices and regularizers themselves. (By construction, selecting any action changes the states of all regularizers; therefore, the compatibility graph must contain directed edges from each action vertex to every regularizer vertex.) To generate the action feedback graph, we iterate over all possible directed edges between action vertices. For each candidate edge, we sample a uniform random number $u \sim (0, 1)$ and include the edge in the graph if $u > 0.2$; otherwise, it is excluded.

Figure 2 compares the dynamic regret (blue) and weak regret (green) across the three experimental cases, with one subplot per case. Figure 3 plots the corresponding objective values ($\mathrm{OPT}_D$, $\mathrm{OPT}_W$, computed via CVXPY (Diamond & Boyd, 2016)) and the cumulative reward $R(\mathrm{Alg})$ achieved by Algorithm 1 for each case. Results are averaged over 30 trials, with shaded regions indicating $\pm$ one standard deviation; subplots in the same row share a common y-axis.

From the figures above, the baseline performs worse because it does not utilize the graph structure. The time-varying graph performs slightly better than the time-invariant graph, potentially due to the varying topology providing more informative feedback over time. The optimal values of $\mathrm{OPT}_D$ and $\mathrm{OPT}_W$ were computed using CVXPY; each corresponds to an integer programming problem. The dynamic benchmark $\mathrm{OPT}_D$ is harder to solve because it involves a larger set of integer variables: while $\mathrm{OPT}_W$ requires only 5 variables (one per action), $\mathrm{OPT}_D$ requires $5 \times T$ variables (one per action per round). Consequently, the computed values for $\mathrm{OPT}_D$ exhibit greater variance in the figure.

The implementation and experimental scripts are publicly available at `https://github.com/Quan-Zhou/graph-feedback-fair-online-learning`.

# 7 Conclusions

In this paper, we have addressed the challenge of bandit-based sequential decision-making under non-stationary fairness constraints, a setting that arises in many real-world resource allocation systems where decisions must be made with limited feedback. Our contributions are twofold. First, we formulated this problem as an online bandit problem with graph-structured partial feedback, explicitly modeling the interplay

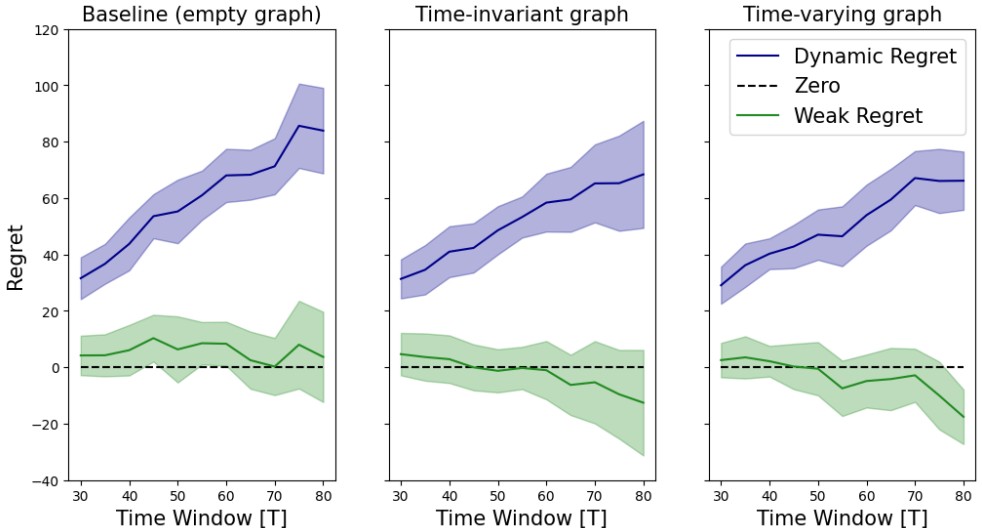

Figure 2: Dynamic (blue) and weak (green) regret of Algorithm 1 under the three experimental configurations. Performance is evaluated over 30 independent trials, each with a randomly sampled batch of income vectors. The curves represent mean regret across trials, with shaded error bands corresponding to ± one standard deviation.

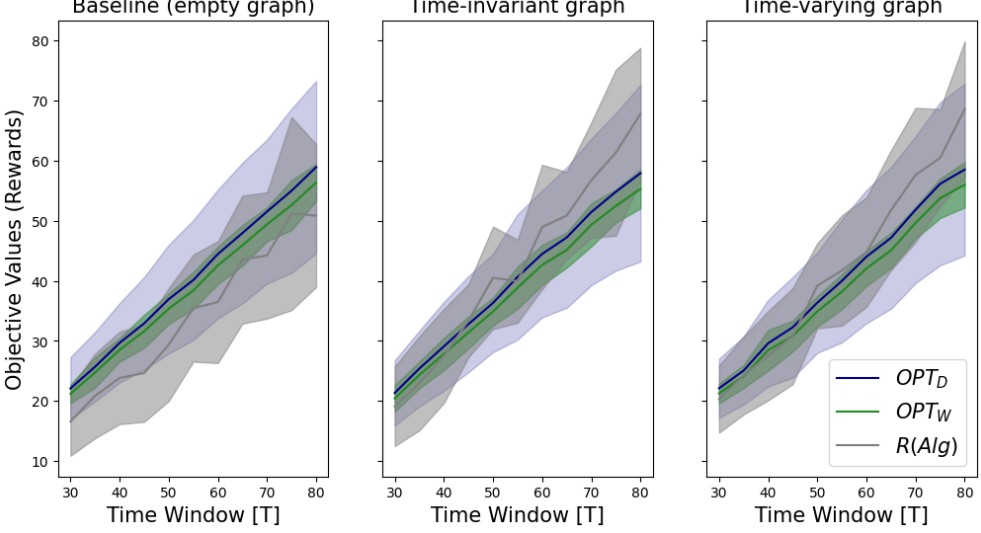

Figure 3: Rewards achieved by Algorithm 1 in the three experimental cases, alongside the computed optimal values $\text{OPT}_D$ (blue) and $\text{OPT}_W$ (green) via the CVXPY package (Diamond & Boyd, 2016). Results are averaged over 30 trials, each using a randomly selected batch of revenue vectors. Curves show means, with shaded bands indicating ± one standard deviation.

between actions within the bandit framework. Second, in our approach, fairness regularizers are represented as additional vertices in the feedback graph, which traditionally only encoded relationships among actions. This formulation enables the bandit decision-maker to learn from structured feedback while dynamically balancing reward objectives with multiple fairness goals that may evolve over time.

**Acknowledgments**

This work has received funding from the European Union's Horizon Europe research and innovation programme under grant agreement No. 101070568. This work was also supported by Innovate UK under the Horizon Europe Guarantee; UKRI Reference Number: 10040569 (Human-Compatible Artificial Intelligence with Guarantees (AutoFair)). J.M. also acknowledges the support of National Recovery Plan funded project MPO 60273/24/21300/21000 CEDMO 2.0 NPO. R.S. was in part supported by the IOTA Foundation.

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

## A    Proof of Theorem 1

*Proof.* We want to show that with learning rate $\eta \leq 1/(3|A|)$ sufficiently small such that $\beta \leq 1/4$, with probability at least $1 - \delta$, we have that the weak regret of Algorithm 1 is upper bounded by equation 8, where $\tilde{\mathcal{O}}$ hides only numerical constants and factors logarithmic in $|A|$ and $1/\eta$.

$$
\sqrt{5\log(\frac{5}{\delta}) \sum_{t\in[T]} \mathrm{mas}(\mathcal{G}^t)} + 12\eta\sqrt{\frac{\log(5|A|/\eta)}{\log|A|} \sum_{t\in[T]} \mathrm{mas}(\mathcal{G}^t)}
$$
$$
+ \tilde{\mathcal{O}}\left(1 + \sqrt{T}\eta + T\eta^2\right)\left(\max_{t\in[T]} \mathrm{mas}^2(\mathcal{G}^t)\right) + \frac{2\log(5|A|/\delta)}{\eta},
\tag{8}
$$

To prove this, we refer to Theorem 9 in Alon et al. (2017). (9-10) use the definition of $W^t, \omega^{(a,t)}, p^{(a,t)}$ in Algorithm 1. equation 11 uses inequality $\exp(x) \leq 1 + x + x^2$.

$$\frac{W^{t+1}}{W^t} = \sum_{a \in A} \frac{\omega^{(a,t+1)}}{W^t} = \sum_{a \in A} \frac{\omega^{(a,t)}}{W^t} \exp(\eta \hat{r}^{(a,t)}) \tag{9}$$

$$= \sum_{a \in A} \frac{p^{(a,t)} - \gamma^t \xi^{(a,t)}}{1 - \gamma^t} \exp(\eta \hat{r}^{(a,t)}) \tag{10}$$

$$\leq \sum_{a \in A} \frac{p^{(a,t)} - \gamma^t \xi^{(a,t)}}{1 - \gamma^t} \left( 1 + \eta \hat{r}^{(a,t)} + (\eta \hat{r}^{(a,t)})^2 \right) \tag{11}$$

$$\leq 1 + \frac{\eta}{1 - \gamma^t} \sum_{a \in A} \left( p^{(a,t)} \hat{r}^{(a,t)} + \eta p^{(a,t)} (\hat{r}^{(a,t)})^2 \right). \tag{12}$$

equation 13 uses equation 12 and inequality $\ln(x) \leq x - 1$.

$$\ln \left( \frac{W^{T+1}}{W^1} \right) = \sum_{t \in [T]} \ln \left( \frac{W^{t+1}}{W^t} \right) \leq$$
$$\sum_{t \in [T]} \sum_{a \in A} \frac{\eta}{1 - \gamma^t} \left( p^{(a,t)} \hat{r}^{(a,t)} + \eta p^{(a,t)} (\hat{r}^{(a,t)})^2 \right). \tag{13}$$

For a fixed single action vertex $a$, we have

$$\ln \left( \frac{W^{T+1}}{W^1} \right) \geq \ln \left( \frac{\omega^{(a,T+1)}}{W^1} \right) =$$
$$\ln \left( \frac{\omega^{(a,1)} \exp(\eta \sum_{t \in [T]} \hat{r}^{(a,t)})}{|A|} \right) = \eta \sum_{t \in [T]} \hat{r}^{(a,t)} - \ln|A|. \tag{14}$$

From Azuma's inequality, Chernoff bound and Freedman's inequality, we have the upper bound of regret with probability at least $1 - \delta$ in equation 15. Then, equation 16 is obtained by combining (13-14). By substituting condition $\beta = \tilde{\mathcal{O}}(\eta), \gamma^t = \tilde{\mathcal{O}}(\eta \mathrm{mas}(\mathcal{G}^t)) \in [\eta, 1/2]$, we get the upper bound in equation 8, where $\tilde{\mathcal{O}}$ ignores factors depending logarithmically on $|A|$ and $1/\delta$.

$$\sum_{t \in [T]} r^{(a,t)} - r^{(a^t,t)}$$

$$\leq \left( \sum_{t \in [T]} \hat{r}^{(a,t)} - \sum_{t \in [T]} \sum_{a \in A} p^{(a,t)} \hat{r}^{(a,t)} \right) + \frac{\ln(k/\delta)}{\beta} + \sqrt{\frac{T \ln(|A|/\delta)}{2}} \tag{15}$$

$$+ \sqrt{2 \ln(1/\delta) \sum_{t \in [T]} \mathrm{mas}(\mathcal{G}^t)} + \beta \sum_{t \in [T]} \mathrm{mas}(\mathcal{G}^t) + \tilde{\mathcal{O}} \left( \max_{t \in [T]} \mathrm{mas}(\mathcal{G}^t) \right)$$

$$\leq \frac{\eta}{1 - \max_{t \in [T]} \gamma^t} \sum_{t \in [T]} \sum_{a \in A} \left( p^{(a,t)} (\hat{r}^{(a,t)})^2 + \gamma^t p^{(a,t)} \hat{r}^{(a,t)} \right) + \frac{\ln|A|}{\eta} \tag{16}$$

$$+ \frac{\ln(k/\delta)}{\beta} + \sqrt{\frac{T \ln(|A|/\delta)}{2}} + \sqrt{2 \ln(1/\delta) \sum_{t \in [T]} \mathrm{mas}(\mathcal{G}^t)} + \beta \sum_{t \in [T]} \mathrm{mas}(\mathcal{G}^t) + \tilde{\mathcal{O}} \left( \max_{t \in [T]} \mathrm{mas}(\mathcal{G}^t) \right).$$

$$\square$$

