# OpenReview forum: "Online Learning with Multiple Fairness Regularizers via Graph-Structured Feedback"
_TMLR — Accepted by TMLR_

### Review · Reviewer_SGyL · 2025-11-17

**Summary Of Contributions:**

The paper presents a sequential decision-making framework for exploring tradeoffs between multiple fairness measures. The proposed approach extends the graph-structured bandit problem by adding extra graph nodes that represent fairness measures. Within this context, the authors focus on the case of limited graph-structured feedback, where the fairness measure state updates corresponding to each selected action are encoded by edges connecting that action node and the corresponding fairness measure nodes.

**Audience:**

Yes

**Audience Explanation:**

The idea of explicitly encoding the connections between fairness measures and actions in a graph-structured framework seems quite interesting. I suspect it could serve as a useful basis for developing future algorithms related to fairness tradeoffs.

**Broader Impact Concerns:**

No concerns.

**Claims And Evidence:**

No

**Claims Explanation:**

I find it difficult to evaluate this work because it is not very clear about what claims it is trying to make. However, there are enough deficiencies in various areas to raise questions about the value of the proposed approach.

The abstract justifies the basic idea of considering multiple fairness notions and provides high-level details of the proposed approach. The conclusion is a single sentence that reiterates a high-level description of the proposed framework. Neither of these sections provide information or claims about the novelty of the proposed method compared to prior works.

The introduction provides elaboration on the intended use case of the approach, but there is no explicit list of primary contributions.

The proposed approach is evaluated on a resource allocation dataset, but the paper does not compare its results to any baseline approach. There is also no discussion of the results, making their relevance unclear and making it hard to tell how they support the paper's claims.

The appendix provides a proof of an upper bound on regret for the proposed approach, which is appreciated. However, the proof goes nearly unmentioned throughout the rest of the work, and the authors never attempt to leverage the derived upper bound to justify any advantage of the proposed approach over prior works.

Overall, I do not think the work does a good job of clearly articulating the claims it is trying to make, and I do not believe that the evidence provided by the work is sufficient in its current state to support the claims it should be making.

**Requested Changes:**

Necessary changes:

+ Update and expand the abstract and introduction to include a clearer set of claims regarding the unique value provided by your framework compared to prior works.

+ Add a "main contributions" list to the end of the introduction section.

+ Provide a discussion of the results shown in Figures 2 and 3. How do these results support your paper's claims?

Optional changes that would improve the work:

+ Update Figures 2 and 3 with some comparison to a baseline approach. If no baseline exists or if it doesn't make sense here to compare to a baseline, add a paragraph for the reader that explains why.

+ Replace the "Case I" and "Case II" chart titles with "Time-Invariant Graph" and "Time-Varying Graph". You have enough space to make those graphs wider to fit a clearer title.

+ If it makes sense, you should use the derived regret upper bound from the proof of Theorem 1 to help justify claims about the proposed framework.

+ Update and expand the conclusions section accordingly with the rest of the changes you make.

---

### Review · Reviewer_twZu · 2025-11-20

**Summary Of Contributions:**

This work proposes a framework for tackling multiple possibly conflicting fairness objectives in a sequential decision-making problem. The authors formulate actions and fairness regularizers as nodes in a graph-structured model, and maintain the balance between fairness and performance under limited bandit feedback and time-varying network topology. The authors adapt and extend existing graph-structured bandit algorithms (notably Alon et al., 2017) and provides theoretical regret guarantees as well as illustrative numerical experiments on a political advertising scenario.

**Additional Comments:**

Minor comment:

In Equation (3), is the state dependent on the specific regularizer or not ($s^t$ or $s^{(f,t)}$)?

**Audience:**

Yes

**Audience Explanation:**

The need of maintaining good trade-offs among multiple potentially conflicting fairness measures does exists in many practical applications. Therefore, it is beneficial to see how this need can be formulated from different perspectives.

**Claims And Evidence:**

No

**Claims Explanation:**

Although the paper is understandable, the structure/organization of the presentation could be improved. Specifically, in the abstract, the authors mentioned the learning of time-varying convexifications of multiple fairness measures, but there are no clear mathematical formulations or discussion regarding that. There are also some technical parts which need further clarification (please see the major comments).

**Requested Changes:**

Major comments:

  1. It seems that the proposed algorithm heavily reuses existing algorithms in the literature (e.g., Exp3-SET or the one proposed by Alon et al., 2017). The reviewer wonders whether there are any new insights on the algorithmic side, or if the method essentially follows the same structure with a new interpretation in the context of fairness.

2. The result in Theorem 1 does not fully characterize how fairness-related quantities (e.g., correlations among fairness regularizers) affect the regret bounds or learning dynamics. The regret guarantee appears to be a direct application of existing graph-bandit theory, without providing any fairness-specific insights.

3. The empirical evaluation is limited to a small-scale synthetic example. The number of trials is too small to yield any statistically significant conclusions, and from the simulation results, the reviewer cannot verify how the performance depends on the parameters appearing in the regret bound (e.g., the network structure).

4. In the abstract, the authors highlight the learning of time-varying convexifications (i.e., the relative weights) of multiple fairness measures. However, this idea is not clearly or rigorously formalized in the paper. Further clarification is needed.

---

### Review · Reviewer_y5ii · 2025-12-09

**Summary Of Contributions:**

The authors offer a broad overview of the problems of fairness in machine-learning systems in general and a good high-level overview of graph-structured bandits. They show how the existing framework of graph-structured bandits can be used for some decision-making that considers fairness issues.

**Audience:**

Yes

**Audience Explanation:**

The paper is about a specific field of machine learning (bandit algorithms) and uses the lens of fairness, which ought to see more light. It explains how existing algorithms could be used for learning, decision-making in the bandit setting (with a hidden state), and fairness optimisation.

**Broader Impact Concerns:**

Section not present and not needed.

**Claims And Evidence:**

No

**Claims Explanation:**

I have trouble identifying the expected contribution from the authors in the field, apart from a modelling framework. Based on the beginning of the submission (what is closest to TMLR's "claims", in my opinion), I would have expected a much deeper analysis (details below).

More specifically, what is "time-varying convexification"? The term is not really explained in the manuscript, which seems like an overreaching claim for the current contents.

**Requested Changes:**

Major comments:
- Your introduction is very generic and broad and clearly underlines the need to include fairness. However, the proposed article focuses on fairness in the specific case some stakeholder wants to make decisions with fairness in mind with the strong constraint that the decisions can be made by bandits. This is a very strong assumption and limitation of your work, while it is not as clearly indicated in the title or in the article itself (for instance, you frequently refer to "sequential decision-making", and bandits are a very limited way to perform sequential decision-making). In my opinion, this limitation should be at least hinted at in the title, in the abstract, and in the first two sections.
- What is the main contribution of the article? Based on the abstract and first few sections alone, I would have expected a new algorithm that works in the described setting or more detailed experiments of existing algorithms in the setting you study or a more comprehensive numerical or theoretical study of existing algorithms. Instead, you offer a small theoretical study of a single algorithm that was the first one introduced in your literature survey (I couldn't find earlier attempts) with a very brief numerical experiment of the same algorithm. Your theoretical study focuses on your weak regret, a metric whose limitations you very well highlight in your manuscript.

Minor comments (in order of appearance in the submitted PDF file):
- Section 2.1: "fairness under unawarenes" is not explained, while it would be beneficial to the manuscript to have a brief explanation (one sentence could be enough).
- Section 2.4: I find it surprising that you do not cite prior work on graph-structured bandits in the second paragraph, like your Mannor & Shamir (2011).
- Table 1: having "Algorithm 1", from Cohen et al. (2016), induces some unneeded confusion with your Algorithm 1 (ELP). I would suggest calling this "Cohen algorithm" or "Cohen-Hazan-Koren algorithm" (or CHK in the interest of space) to reduce the potential to mistake the reference's algorithm with your own.
- Section 3: "The notation a t −→ a′ indicates an edge connecting action a to a′", what is the role of t in this notation?
- Section 3, formula 4: wouldn't it be clearer to write $\sum_t \max_{a\in V} r^t\left(a\right)$ ?
- Theorem 1: this is the first use of the notation mas(G) outside of a caption (Table 1, many pages before). Please consider summarising your notations in a dedicated space, at least for the notations used more generally in the submission (e.g., χ does not really need to be defined outside the caption of Table 1). It would be easier to follow, rather than to look for definitions that are scattered throughout the manuscript.
- Section 5: in the sentence starting with "The revenue of the platform, is assumed", is this the only use of $\dot s$? I didn't see a definition of it.

---

### Author Response · Authors · 2025-12-25
**Paper6264 Authors' Response**

***We would like to take this opportunity to express our sincere thanks to the Editors and Reviewers for their insightful feedback. We found the reviews invaluable, and they have greatly strengthened our work. We have uploaded the revised draft with all changes highlighted in blue for easy reference. Our detailed point-by-point response follows below.***


**To Reviewer y5ii**

What is "time-varying convexification"?
>Thank you for this comment. Our contribution is better described as embedding time-varying fairness constraints as regularizer vertices in a bandit feedback graph. We have renamed the paper to “Online Learning with Multiple Fairness Regularizers via Graph-Structured Feedback.”

Hint at limitations in title/abstract.
> We have specified our focus on bandit-based sequential decision-making in the abstract and first paragraph (see blue text).

What is the main contribution?
> We have added a contribution paragraph at the end of Section 1 (see blue text).

Explain "fairness under unawareness."
>We have added an explanation (see blue text in Section 2.1).

Cite prior work on graph-structured bandits.
> We have added the citation (see blue text in Section 2.4).

Table 1 confusion.
> We have renamed the reference to “CHK” (see blue text in Table 1).

Notation a t → a'.
> We have rewritten and explained the notation in Section 3 (see blue text at the head of page 5).

Section 3. Formula 4.
>Thank you for this suggestion.  We have updated it as Equation (3).

Summarise notations.
>We have added a notation list in Table 2.

> the use of \dot s.
We apologize for the typo and have corrected the notation (see orange text on page 9).


**To Reviewer twZu**

New algorithmic insight?
> The update steps follow ELP. The key insight is our graph construction: we introduce fairness regularizer vertices with edges encoding each action's influence on fairness measures.

fairness-specific insight.
> Fairness constraints change the feedback topology. Adding regularizer vertices introduces new edges (action→regularizer) that affect observable feedback.

Limited empirical evaluation.
> We have increased the number of trials from 5 to 30 per parameter set.

"Time-varying convexifications" not formalized.
> Thank you for this comment. We have rewritten the title and abstract for clarity.

State dependency in Eq. (3).
>We apologize for the typo. We have corrected it to s^(f,t) (see blue text in Equation (2)).


**To Reviewer SGyL**

Add contributions list.
> We have added this at the end of Section 1 (see blue text).

Discuss Figures 2 and 3.
> We have added discussion in the last two paragraphs of Section 6.

Update figures with a baseline.
> We added a baseline using an empty graph.

Replace "Case I" and "Case II."
> We have replaced the titles "Case I" and "Case II" in the figures with descriptive labels from the new list on page 10.

Use the derived regret bound.
> We mention the proof right after Theorem 1 and added explanation.

Expand conclusions.
> We have made the conclusion more detailed and stressed the contribution.

---

### Author Response · Authors · 2025-12-28
**Re: Minor changes**

Dear Reviewer y5ii,

Thank you for your careful review. We have uploaded the revised draft. Corrections have been made:

1. Changed to "...and, in our setting, must be learned..."

2. Corrected "The the size" to "The size..."

---

### Decision · Action_Editor_ZukZ · 2026-02-18

**Recommendation:** Accept as is

**Additional Comments:**

Congratulations!  Please prepare your final camera ready version, making sure to insert the author names and other required publication information and turn off the text coloring added for review purposes.

**Audience:**

Yes

**Audience Explanation:**

All three reviewers unanimously agree at the topics, bandits and fairness, are of interest to the TMLR community and that the paper meets the standard of having result of interest to some of them.

**Claims And Evidence:**

Yes

**Claims Explanation:**

The reviewers raised a number of concerns about the clarity and justification of the claims in the paper and the authors responded to these with substantial revisions to address them.  The major changes included:
 - Rewriting the abstract to make sure the claims in it are justified
 - Adding an explicit discussion of the contributions to the introduction
 - Clarifying details and notation in the model
 - Better connecting the theory to the claims
 - Increasing the scale of the experiments
 - Revising the conclusion to rearticulate the main claims

After these changes, two of the reviewers were satisfied that this criterion was satisfied while a third was not.  I reviewed the reviewer concerns and the changes and am satisfied that they are adequate to meet the bar for TMLR.  Specifically, in the revised version the claims focus on the introduction of a new formulation and algorithms that are appropriate for this setting.  The paper provides sufficient evidence that they are appropriate and potentially interesting, which is all that is needed to meet TMLR's standard in terms of justifying these claims.